# Talk2Me: High-Fidelity and Controllable Audio-Driven Avatars with Gaussian Splatting

## Abstract

Audio-driven avatars are increasingly employed in online meetings, virtual humans, gaming, and film production. However, existing approaches suffer from technical limitations, including low visual fidelity (e.g., facial collapse, detail loss) and limited controllability in expression and motion, such as inaccurate lip synchronization and unnatural head motion. Besides, most existing methods lack explicit modeling of the correlation between facial expressions and head pose dynamics, which compromises realism. To address these challenges, we propose Talk2Me, a high-fidelity, expressive, and controllable audio-driven framework comprising three core modules. Firstly, we enhance 3D Gaussian Splatting (3DGS) with Learnable Positional Encoding (LPE) and a modified Region-Weighted Mechanism to mitigate facial collapse and preserve fine details. Secondly, an Expression Generator (EG) with an Audio-Expression Temporal Fusion (AETF) module models the temporal relationship between audio and expression features, enabling accurate lip-sync and natural expression transitions. Thirdly, a Retrieval-Based Pose Generator (RBPG) explicitly captures the coupling between expressions and pose dynamics, with a Pose Refiner (PR) enhancing the naturalness and continuity of head motion. We further construct a Mandarin monocular video dataset featuring diverse identities to evaluate cross-lingual generalization. Experiments demonstrate that Talk2Me outperforms state-of-the-art methods in visual quality, synchronization accuracy, and motion naturalness.

## 1 Introduction

Audio-driven talking avatars have received increasing attention from both academia and industry. This task involves cross-modal synthesis, where visual facial animations must be temporally aligned with input audio. Such audio-driven avatars show strong potential in applications like virtual conferencing, gaming, and film production (Li et al., 2023; Peng et al., 2023; Cho et al., 2024; Peng et al., 2024). By bridging human interaction and digital media, this technology plays a central role in immersive experiences and intelligent agents.

Despite recent advances, audio-driven avatar generation continues to face two fundamental challenges: achieving high visual fidelity and ensuring controllable facial and head dynamics. Methods based on Generative Adversarial Networks (GANs) (Guan et al., 2023; Wang et al., 2023; Zhang et al., 2023b; Zhong et al., 2023) synthesize talking avatars conditioned on audio or landmarks, but often suffer from identity inconsistency and inter-frame jitter (Peng et al., 2024), undermining overall realism and temporal stability. Methods based on Neural Radiance Fields (NeRFs) (Mildenhall et al., 2021; Guo et al., 2021; Shen et al., 2022; Yao et al., 2022) improve structural modeling through volumetric rendering, yet frequently exhibit facial collapse, loss of fine-grained details, and poor lip-speech synchronization. Recently, 3D Gaussian Splatting (3DGS) (Kerbl et al., 2023) has emerged as a compelling alternative due to its efficient rendering and strong spatial modeling capabilities. However, existing 3DGS-based methods (Cho et al., 2024; He et al., 2024; Li et al., 2024) still struggle with expression jitter, temporal incoherence, and imprecise lip synchronization, limiting both fidelity and controllability.

To overcome low fidelity and limited controllability in audio-driven avatars, we revisit key limitations of existing methods. NeRF- and 3DGS-based approaches rely heavily on traditional sinusoidal positional encoding, which fails to capture local spatial variations and causes facial collapse with loss of fine details. More broadly, GAN-, NeRF-, and 3DGS-based methods suffer from temporal misalignment between audio and expression features, leading to lip-sync errors and discontinuous expressions, as well as the independent treatment of expression and head pose, which produces rigid motion and poor controllability. Inspired by these limitations, we introduce Talk2Me, a 3DGS-based framework tailored to enhance visual fidelity and enable controllable facial and head dynamics. Built upon 3DGS, Talk2Me incorporates several targeted modules to address the identified challenges.

Facial collapse and the loss of fine-grained details are two key obstacles to high-fidelity avatar generation. We address these issues by reforming the avatar modeling framework with Learnable Positional Encoding (LPE) and a modified Region-Weighted Mechanism, which more effectively capture spatial relationships among Gaussian primitives and improve structural consistency and detail preservation. To further improve controllability over expressive dynamics, we incorporate the Eye Aspect Ratio (EAR) (Dewi et al., 2022) feature into the expression representation, enabling fine-grained modulation of blinking. It is also worth noting that the inherent modeling capability of 3DGS naturally ensures identity consistency throughout the animation process.

For expression controllability, we introduce an Expression Generator (EG) equipped with an Audio-Expression Temporal Fusion (AETF) module. This component jointly models audio and expression features across time, enabling accurate lip synchronization and smooth expression transitions. A region-aware attention mechanism further refines lip and eye details, enriching facial features and improving lip-sync precision.

For pose controllability, we introduce a Retrieval-Based Pose Generator (RBPG) alongside a dedicated Pose Refiner (PR), which jointly generate natural and expressive head movements. PR takes the retrieved pose, expression, and audio features as input, performing structured cross-modal fusion and temporal modeling via multi-layer transformers. By capturing the intrinsic correlation between expressions and head motion and optimizing temporal dynamics, our method ensures coherent and lifelike head behavior. Furthermore, to assess cross-lingual generalization, we curate a Mandarin video dataset featuring diverse identities and speech content.

Leveraging the above strategies, Talk2Me achieves high-fidelity, expressive, and controllable avatar synthesis, with precise lip synchronization, natural head motion, and robust facial detail preservation. Evaluations on both English and Mandarin datasets confirm its superiority over existing methods in generation quality, synchronization accuracy, motion coherence, and expression controllability.

Our main contributions are summarized as follows:

- We present Talk2Me, an audio-driven avatar framework designed to improve both visual fidelity and motion controllability in expressive talking head synthesis.
- We enhance 3D Gaussian Splatting with a Learnable Positional Encoding (LPE) and a modified Region-Weighted Mechanism, effectively addressing facial collapse and enabling fine-grained expression control.
- We propose an Expression Generator (EG) for expressive and controllable facial expression synthesis, and a Retrieval-Based Pose Generator (RBPG) to model expression–pose correlation, enhancing head motion naturalness and controllability.
- Extensive experiments on both English and Mandarin corpus show that Talk2Me delivers more faithful and controllable talking avatars than existing methods.

## 2 RELATED WORK

### GAN-BASED METHODS.

Early works (Chen et al., 2018; Prajwal et al., 2020; Sun et al., 2022; Guan et al., 2023; Wang et al., 2023; Zhong et al., 2023) focus on synthesizing only the mouth region. Although they achieve accurate lip synchronization, they often neglect identity preservation. Later works (Chen et al., 2019; Das et al., 2020) expand to full-face generation to address visual discontinuities, but identity

drift and inter-frame jitter still remain. More recent efforts (Song et al., 2022; KR et al., 2019; Zhou et al., 2020; Lu et al., 2021; Ji et al., 2022; Zhang et al., 2023a; 2021a; Zhou et al., 2021) incorporate head pose, blinking, and emotion to improve expressiveness. However, due to frame-wise generation and the absence of temporal constraints, these models still struggle to ensure visual fidelity and controllable motion. In contrast, Talk2Me leverages the 3DGS representation to ensure identity consistency, high-fidelity rendering, and temporally coherent control.

NeRF-based Methods.

NeRF-based methods improve identity consistency and spatial coherence over 2D GANs, enabling higher-quality talking avatars. Yet, early audio-driven NeRFs often suffer from facial collapse and detail loss, degrading fidelity and lip synchronization. Subsequent works attempt to mitigate these issues—AD-NeRF (Guo et al., 2021) drives NeRFs directly from audio, DFA-NeRF (Yao et al., 2022) decouples facial attributes, and SSP-NeRF (Liu et al., 2022) introduces semantic-aware modeling. To enhance efficiency, RAD-NeRF (Tang et al., 2025) and ER-NeRF (Li et al., 2023) accelerate training and rendering, while SyncTalk (Peng et al., 2024) emphasizes precise lip–pose synchronization. Despite these advances, challenges remain in reconstructing high-frequency facial details (especially lips), avoiding subtle collapse, and achieving fine-grained controllability. In contrast, Talk2Me builds on 3DGS with LPE and couples expression representation with EAR to ensure high fidelity and controllable expression dynamics.

3DGS-based Methods.

Recent advances in 3D Gaussian Splatting (3DGS) have demonstrated strong capabilities in scene modeling and efficient rendering, making it increasingly popular for audio-driven avatar synthesis. Pioneering works such as EmoTalk3D (He et al., 2024), GaussianTalker Cho et al. (2024), and TalkingGaussian (Li et al., 2024) have begun to explore this direction, with a particular focus on accelerating the rendering process. However, these methods often lack explicit temporal modeling between audio signals and expression features, leading to common issues such as expression jitter, lip-sync mismatches, and overall temporal incoherence. These shortcomings ultimately limit the realism and controllability of the generated avatars.

Unlike previous 3DGS-based methods, Talk2Me introduces EG to generate temporally coherent and controllable facial expressions while improving lip-sync accuracy. In addition, it incorporates RBPG to explicitly model the correlation between facial expression and head pose, leading to more natural and controllable head movements.

## 3 Preliminary

### 3.1 3D Gaussian Splatting.

3D Gaussian Splatting (3DGS) is a real-time rendering method that models a scene with a set of anisotropic 3D Gaussian primitives. Each primitive is parameterized by a spatial center $\boldsymbol{\mu} \in \mathbb{R}^3$, scale $\mathbf{s}$, rotation $\mathbf{q}$, feature vector $\mathbf{z}$, color $\mathbf{c}$, and opacity $\boldsymbol{\alpha}$. These primitives collectively encode the geometry, appearance, and transparency of the scene.

Unlike NeRF-based methods relying on dense ray marching, 3DGS treats each Gaussian as an elliptical blob projected onto the screen and blended via differentiable rasterization, achieving high-quality, real-time rendering.

Given $N$ Gaussians $\{G_i\}_{i=1}^N$, the pixel color is obtained by alpha-compositing weighted Gaussians:

$$C = \sum_{i=1}^{N} w_i \cdot \mathbf{c}_i, \quad \text{where} \quad w_i = \alpha_i \cdot T_i \tag{1}$$

where $T_i$ is the transmittance term accounting for occlusion and depth.

3DGS supports end-to-end optimization of geometry and appearance. However, directly applying it to talking head synthesis can cause facial collapse or detail loss under large deformations, motivating our improvements.

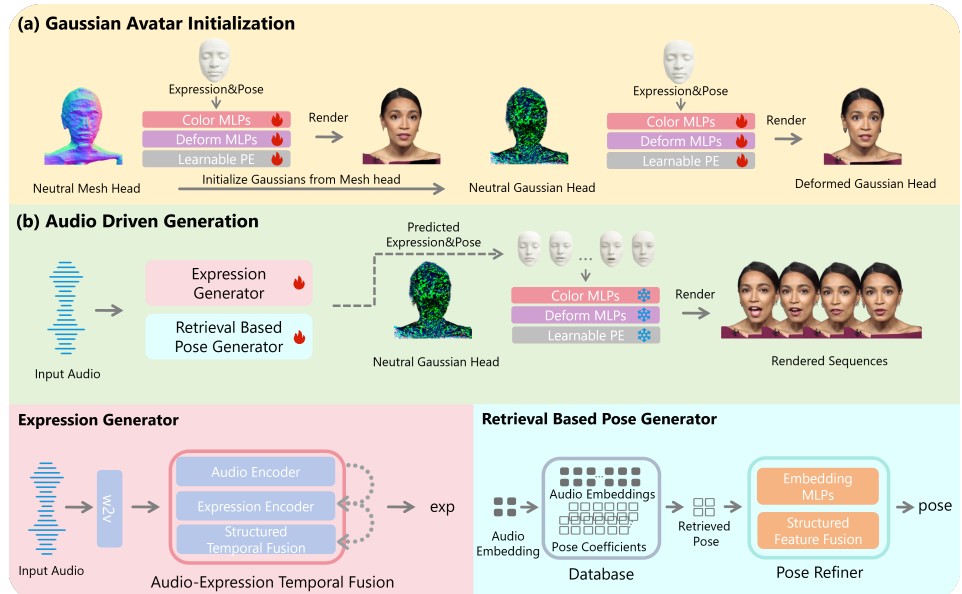

Figure 1: **Overview of Talk2Me.** Our framework has two stages: (a) Gaussian Avatar Initialization, where a mesh-based talking head initializes a 3D Gaussian avatar; (b) Audio-Driven Generation, where the Expression Generator (EG) and Retrieval-Based Pose Generator (RBPG) predict expressions and poses from audio to produce expressive, temporally coherent renderings.

## 4 METHOD

### 4.1 OVERVIEW

In this section, we present the proposed *Talk2Me*, as illustrated in Figure 1. It comprises two stages: (a) *Gaussian Avatar Initialization*, which builds a 3D Gaussian head from a mesh-based avatar, and (b) *Audio-Driven Generation*, where the Expression Generator (EG) and Retrieval-Based Pose Generator (RBPG) predict expression and pose from input audio to drive avatar animation. The internal designs of EG and RBPG are shown at the bottom of Figure 1, and we describe each module in detail below. For more detailed model designs, please refer to the Appendix A.3.

### 4.2 GAUSSIAN AVATAR INITIALIZATION

Following HHAvatar (Liao et al., 2023), we use DMTet to extract a neutral mesh head, which serves as the foundation for initializing 3D Gaussian primitives in a canonical space.

#### 4.2.1 REGION-WEIGHTED MECHANISM.

We aim to build an animatable Gaussian Avatar driven by expression $\mathbf{e}$ and head pose $\mathbf{p}$. Given audio, we predict temporally aligned $\mathbf{e}$ and $\mathbf{p}$ to deform Gaussian primitives for expressive, high-fidelity animation.

To balance local expressivity and global stability, we adapt the region-weighted mechanism from HHAvatar (Liao et al., 2023), integrating expression features. Facial regions near landmarks (e.g., mouth, eyes) are assigned higher expression weights

$$w_{\mathrm{exp}} = \mathrm{clamp}\left(\frac{d_{\mathrm{far}} - d}{d_{\mathrm{far}} - d_{\mathrm{near}}}, 0, 1\right), \tag{2}$$

where $d$ is the distance to the nearest neutral landmark, and $w_{\mathrm{pose}} = 1 - w_{\mathrm{exp}}$. Thus, expressive regions mainly follow $\mathbf{e}$ (augmented with EAR for ocular details), while peripheral regions follow $\mathbf{p}$ for coherent global motion.

We implement controllable deformation using an MLP-based network $\phi_D$ that takes primitive features $\mathbf{z}$, expression $\mathbf{e}$, pose $\mathbf{p}$, and ocular-aware features $\boldsymbol{\beta}$, and outputs attribute offsets:

$$(\boldsymbol{\mu}', \mathbf{s}', \mathbf{q}', \mathbf{c}', \boldsymbol{\alpha}') = \phi_D(\boldsymbol{\mu}, \mathbf{s}, \mathbf{q}, \mathbf{c}, \boldsymbol{\alpha}; \mathbf{e}, \mathbf{p}, \mathbf{z}, \boldsymbol{\beta}). \tag{3}$$

The updated primitives are rendered by a differentiable rasterizer:

$$I = \mathcal{R}(\boldsymbol{\mu}', \mathbf{s}', \mathbf{q}', \mathbf{c}', \boldsymbol{\alpha}'), \tag{4}$$

producing temporally aligned, expressive, and controllable Gaussian Avatars.

### 4.2.2 LEARNABLE POSITIONAL ENCODING.

To enhance deformation accuracy in dynamic facial regions, we introduce a hybrid positional encoding that integrates explicit coordinates, sinusoidal features, and a learnable position table. For each Gaussian primitive $i$ with coordinate $\mathbf{x}_i$, we compute a linear coordinate embedding $\mathrm{PC}(\mathbf{x}_i)$ and an index-based sinusoidal encoding

$$S(i) = \big[ \sin(i \cdot (\gamma \odot \mathbf{f}) + \boldsymbol{\phi}), \ \cos(i \cdot (\gamma \odot \mathbf{f}) + \boldsymbol{\phi}) \big], \tag{5}$$

where $\mathbf{f}$ is the frequency band vector, $\gamma$ a learnable global scale, and $\phi$ a learnable phase. A learnable positional table $L(i)$ further models local offsets, and we blend them as:

$$\widetilde{P}(i) = \sigma(\alpha) \, L(i) + \big(1 - \sigma(\alpha)\big) \, S(i). \tag{6}$$

The final feature is:

$$\mathbf{z}_i = \mathrm{PC}(\mathbf{x}_i) + \widetilde{P}(i). \tag{7}$$

This design provides complementary benefits: $\mathrm{PC}(\mathbf{x}_i)$ encodes geometric priors, $S(i)$ captures multiscale frequency information with adaptive band distribution, and $L(i)$ learns local nonlinear deviations. The sigmoid-controlled blending ensures a smooth trade-off between global Fourier priors and local corrections, yielding better modeling of complex deformations in regions such as eyes.

## 4.3 AUDIO-DRIVEN GAUSSIAN AVATAR

To enable high-fidelity and controllable talking head synthesis, we extend the neutral Gaussian Avatar by incorporating audio-driven dynamics. Our framework introduces two key modules: the *Expression Generator* (EG) and the *Retrieval-Based Pose Generator* (RBPG), which respectively predict temporally aligned facial expressions and head poses from audio. These modules collaboratively produce synchronized, expressive, and naturalistic animations.

### 4.3.1 EXPRESSION GENERATOR.

Mapping speech to coherent facial animation is challenging due to the intricate coupling between phonetic content and facial dynamics. Our Expression Generator (EG) addresses this with two key modules: *Audio-Expression Temporal Fusion (AETF)* and a *Region-Aware Attention Mechanism*.

*Audio-Expression Temporal Fusion.* As shown in Figure 1 (bottom-left, pink), AETF comprises an Audio Encoder, an Expression Encoder, and a Structured Temporal Fusion block. The Audio Encoder extracts phonetic and emotional cues from a pretrained Wav2Lip model, while the Expression Encoder derives style features from a reference frame. Both are projected into a shared latent space, where the fusion block models their temporal interplay for context-aware, audio-aligned expressions.

To capture second-order interactions, we propose a *K-product fusion* mechanism. Given audio $\mathbf{a}$, emotion $\mathbf{e}$, and style $\mathbf{s}$ features, their pairwise Kronecker products are projected as

$$\boldsymbol{f}_e = \phi_E(\mathbf{a} \otimes \mathbf{e}, \ \mathbf{a} \otimes \mathbf{s}, \ \mathbf{e} \otimes \mathbf{s}), \tag{8}$$

where $\phi_E$ is a learnable projection. The fused representation is refined via gated concatenation and fed into a Transformer-based temporal encoder, yielding smooth expression trajectories.

*Region-Aware Attention Mechanism.* To improve fidelity in perceptually critical areas, we adopt a region-aware decoding strategy. A global decoder predicts full-face expressions, while a multi-head attention module refines region-specific dynamics (e.g., eyes) over fused temporal features.

Their outputs are merged into a unified representation, ensuring both lips and eyes exhibit precise, synchronized motion.

We further employ a dual-pathway design to balance phoneme responsiveness and contextual expressiveness. The main pathway leverages AETF to produce emotion- and style-aware features, while a parallel direct audio-to-expression stream captures sharp, phoneme-synchronous lip movements. The two are fused and passed through a lightweight temporal smoothing layer, yielding expressive, temporally stable facial animation.

### 4.3.2 RETRIEVAL-BASED POSE GENERATOR.

Most methods neglect the correlation between head motion and facial expression, causing unstable dynamics despite accurate lip-sync. We address this with the *Retrieval-Based Pose Generator (RBPG)*, which grounds pose in real motion data and refines it with expression–audio cues for smooth, speech-synchronized trajectories.

Given a monocular talking video of the target, we build an audio–pose database $\mathcal{D}$ segmented into $N$-frame units. At inference, the input audio is encoded to $\mathbf{a}'$ and matched via cosine similarity to retrieve an initial pose $\hat{\mathbf{p}}$.

*Pose Refiner.* To refine $\hat{\mathbf{p}}$, we fuse it with expression features $\mathbf{e}'$ via a Kronecker-product interaction:

$$\boldsymbol{f}_{\mathrm{p}} = \phi_{\mathrm{P}}(\hat{\mathbf{p}} \otimes \mathbf{e}'), \tag{9}$$

where $\phi_{\mathrm{P}}$ is a learnable fusion network. A cross-attention block then incorporates audio $\mathbf{a}'$ to align pose with speech prosody. A Transformer decoder subsequently predicts residual corrections $\Delta\mathbf{p}$, producing the final pose $\mathbf{p} = \hat{\mathbf{p}} + \Delta\mathbf{p}$.

This retrieval–refinement framework yields head motions that are temporally smooth, expression-consistent, and synchronized with speech. The detailed architecture of Pose Refiner is illustrated in Figure 12.

### 4.4 TRAINING DETAILS

*Gaussian Avatar.* To achieve high-fidelity facial rendering, we augment standard 3DGS optimization with additional perceptual and adversarial objectives. In addition to $\mathcal{L}_1$, perceptual loss $\mathcal{L}_p$, and SSIM loss $\mathcal{L}_s$, we include an adversarial loss $\mathcal{L}_a$ to improve realism. The overall objective is:

$$\mathcal{L}_{\mathrm{A}} = \lambda_1\mathcal{L}_1 + \lambda_p\mathcal{L}_p + \lambda_s\mathcal{L}_s + \lambda_a\mathcal{L}_a. \tag{10}$$

*Expression Generator.* EG is trained with a weighted $\mathcal{L}_1$ loss between predicted $\hat{\mathbf{e}}$ and ground truth $\mathbf{e}$, assigning higher weights $\mathbf{w}$ to eye-related coefficients:

$$\mathcal{L}_{\mathrm{E}} = \lambda_e \cdot \mathcal{L}_1(\hat{\mathbf{e}}, \mathbf{e}; \mathbf{w}). \tag{11}$$

*Retrieval-Based Pose Generator.* RBPG learns speech- and expression-conditioned head pose sequences with temporal coherence. A dual-branch discriminator evaluates realism globally and locally. The generator loss combines adversarial, reconstruction, and velocity smoothness terms:

$$\mathcal{L}_{\mathrm{P}} = \lambda_{\mathrm{gan}}\mathcal{L}_{\mathrm{gan}} + \lambda_{\mathrm{rec}}\mathcal{L}_1(\mathbf{p}, \hat{\mathbf{p}}) + \lambda_{\mathrm{vel}}\mathcal{L}_1(\mathbf{v}, \mathbf{0}), \tag{12}$$

where $\mathbf{v}_t = \hat{\mathbf{p}}_t - \hat{\mathbf{p}}_{t-1}$ enforces smooth motion, and $\lambda_{\mathrm{rec}} = 10.0$, $\lambda_{\mathrm{vel}} = 1.0$.

## 5 EXPERIMENTS

### 5.1 EXPERIMENT SETTINGS

#### 5.1.1 DATASET.

We utilize monocular talking videos from the HDTF (Zhang et al., 2021b) dataset for training and evaluation. To test cross-lingual generalization, we additionally collect 25 Mandarin videos from diverse identities. Backgrounds are removed using the method in (Lin et al., 2022) to isolate portrait regions. Following prior work, all videos are center-cropped and resized to $512 \times 512$, with a fixed frame rate of 25 FPS.

### 5.1.2 COMPARISON BASELINES.

We compare with GAN-based methods (Wav2Lip (Prajwal et al., 2020), VideoReTalking (Cheng et al., 2022), IP-LAP (Zhong et al., 2023)), NeRF-based methods (ER-NeRF (Li et al., 2023), SyncTalk (Peng et al., 2024), Real3D-Portrait (Ye et al., 2024b), MimicTalk (Ye et al., 2024a)), and 3DGS-based methods (GaussianTalker (Cho et al., 2024), TalkingGaussian (Li et al., 2024)).

### 5.1.3 IMPLEMENTATION DETAILS.

Our method follows a three-stage training pipeline. We first train a neutral mesh head for 30,000 steps, then use it to initialize the Gaussian Avatar, which is further optimized for 20,000 steps. The Expression Generator (EG) and the generator of the Retrieval-Based Pose Generator (RBPG) are trained jointly for 2,000 epochs with a learning rate of $1 \times 10^{-4}$, while the RBPG discriminator uses a learning rate of $1 \times 10^{-3}$. Training completes within a few hours on a single RTX A6000 GPU.

## 5.2 QUANTITATIVE EVALUATION

### 5.2.1 RECONSTRUCTION QUALITY ASSESSMENT.

Following (Li et al., 2023; Cho et al., 2024), we evaluate identity-specific reconstruction on the test set using the audio from the original video. Metrics include PSNR, LPIPS (Zhang et al., 2018), FID (Heusel et al., 2017), MS-SSIM, and Landmark Distance (LMD), measuring photometric accuracy, perceptual realism, and geometric alignment.

### 5.2.2 AUDIO DRIVEN QUALITY ASSESSMENT.

For audio-driven evaluation, where facial and head motions vary, pixel-level metrics are less reliable. We follow Wav2Lip (Prajwal et al., 2020) and report Lip Sync Error Distance (LSE-D) and Lip Sync Confidence (LSE-C) to assess lip–speech alignment using audio from a different video.

### 5.2.3 EVALUATION RESULTS.

Table 1: **Quantitative evaluation of reconstruction and audio-driven talking head synthesis.** We evaluate the methods using standard metrics for visual quality (PSNR, LPIPS, MS-SSIM, FID, LMD) and audio-visual synchronization (LSE-D, LSE-C). **Bold** and underlined indicate the best and second-best results, respectively.

| Methods | | Reconstruction | | | | | | Audio Driven | |
|---|---|---|---|---|---|---|---|---|---|
| | | PSNR↑ | LPIPS↓ | MS-SSIM↑ | FID↓ | LMD↓ | LSE-D↓ | LSE-C↑ | LSE-D↓ | LSE-C↑ |
| **GAN** | Wav2Lip | 32.565 | 0.027 | 0.986 | 5.685 | 2.755 | 6.673 | 8.922 | 8.819 | 6.689 |
| | VideoReTalking | 32.828 | 0.031 | 0.983 | 5.329 | 2.978 | 6.556 | 8.634 | 8.889 | 6.725 |
| | IP-LAP | 33.107 | 0.024 | **0.991** | 5.791 | 2.789 | 6.420 | 8.941 | 9.498 | 5.502 |
| **NeRF** | ER-NeRF | 30.939 | 0.027 | 0.983 | 9.514 | 2.598 | 6.935 | 8.597 | 10.191 | 5.479 |
| | SyncTalk | **36.449** | 0.023 | 0.962 | 6.243 | **2.265** | 8.434 | 6.910 | 9.140 | 6.109 |
| | Real3dPortrait | 21.709 | 0.090 | 0.895 | 28.871 | 5.038 | 6.945 | 8.364 | 10.442 | 4.619 |
| | MimicTalk | 23.233 | 0.064 | 0.979 | 19.638 | 5.239 | 8.353 | 6.827 | 10.021 | 4.104 |
| **3DGS** | GaussianTalker | 28.217 | 0.048 | 0.981 | 21.549 | 2.306 | 7.797 | 7.863 | 11.841 | 3.016 |
| | TalkingGaussian | 30.944 | 0.027 | 0.987 | 9.266 | 2.592 | 6.939 | 8.568 | 9.572 | 5.835 |
| | **Talk2Me** | 29.489 | **0.018** | 0.976 | **5.082** | 3.437 | **6.300** | **8.993** | **8.757** | **6.729** |

Quantitative results for avatar reconstruction and audio-driven synthesis are shown in Table 1, comparing **Talk2Me** with recent GAN-, NeRF-, and 3DGS-based methods. Under the reconstruction setting, **Talk2Me** achieves state-of-the-art perceptual quality (LPIPS, FID) and the best lip–speech synchronization (LSE-D/C). Within the 3DGS family, it clearly outperforms *GaussianTalker* on perceptual metrics, indicating more realistic rendering and tighter audio–visual alignment, while preserving 3D consistency. Although our method does not lead in PSNR or MS-SSIM, this is expected given our *full-head generation* strategy. Unlike models that only edit the mouth in static frames, we synthesize full-face dynamics and head motion—variations that enhance realism and synchronization but may be penalized by pixel-level metrics. In the audio-driven setting, **Talk2Me** achieves

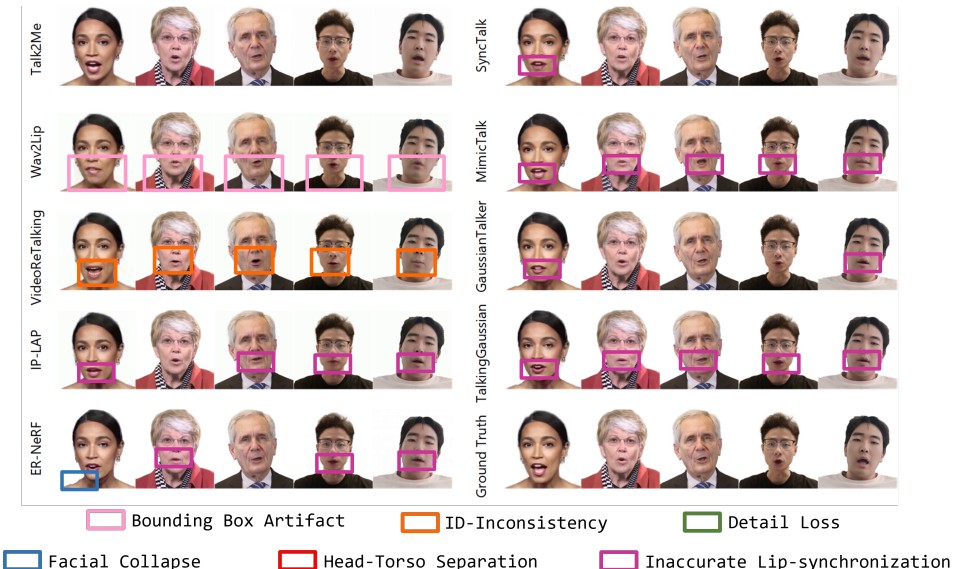

Figure 2: **Qualitative comparison of facial synthesis by different methods.** Compared with GAN-, NeRF-, and 3DGS-based baselines, Talk2Me achieves more accurate lip synchronization, richer facial expressions, and better identity preservation. The results closely match the ground truth (bottom-right) while avoiding artifacts like facial collapse. Please zoom in for details.

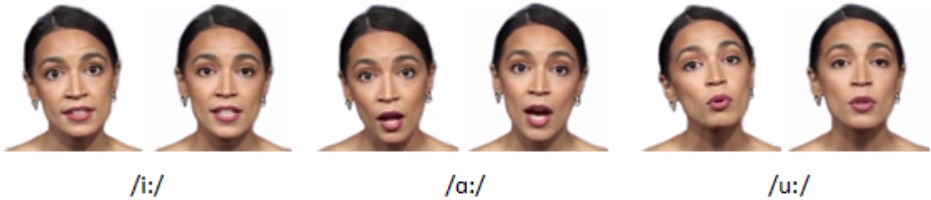

Figure 3: **Head pose diversity with the same audio.** Talk2Me produces natural, diverse head movements while preserving accurate lip-sync. Left: Ground Truth; Right: ours.

the best synchronization across all metrics. Overall, our method delivers high-fidelity, speech-aware animation with fine-grained perceptual detail, controllable expressions, and natural motion.

### 5.3 QUALITATIVE EVALUATION

#### 5.3.1 VISUALIZATION RESULTS.

Figure 2 shows a qualitative comparison across GAN-, NeRF-, and 3DGS-based methods. **Talk2Me** generates expressive facial details, including accurate lip movements and natural head motion, while preserving identity and structural coherence. Unlike GAN-based methods that often distort identity, or NeRF-based ones prone to facial collapse, our method generates fine-grained expression and pose that are well aligned with speech, enabling high-fidelity, controllable, audio driven facial animation. A more detailed comparison is provided in the Appendix A.2.1 and supplementary material.

#### 5.3.2 HEAD POSE GENERATION.

Figure 3 demonstrates **Talk2Me**'s ability to generate natural, diverse head motion from audio while maintaining accurate lip synchronization. Unlike methods that require external driving videos, it predicts temporally coherent head dynamics and expressions directly from speech, enabling fully audio-driven motion.

Table 2: **User study (ratings 1–5) on five aspects. Bold** and underlined indicate the best and second-best results, respectively.

| | Methods | Lip-sync | Expression-sync | Pose-sync | Image quality | Video realness |
|---|---|---|---|---|---|---|
| **GAN** | Wav2Lip | 3.89 | 4.04 | 4.11 | 3.71 | 3.98 |
| | VideoReTalking | 3.72 | 3.85 | 3.97 | 3.72 | 3.66 |
| | IP-LAP | 4.17 | 4.07 | 3.97 | **4.00** | 4.00 |
| **NeRF** | ER-NeRF | 3.18 | 3.26 | 3.23 | 3.08 | 3.18 |
| | SyncTalk | 3.83 | 3.93 | 3.89 | 3.89 | 3.65 |
| | Real3DPortrait | 3.15 | 3.11 | 3.08 | 2.84 | 3.04 |
| | MimicTalk | 3.60 | 3.23 | 3.50 | 3.56 | 3.57 |
| **3DGS** | GaussianTalker | 3.56 | 3.63 | 3.64 | 3.41 | 3.42 |
| | TalkingGaussian | 3.82 | 3.87 | 4.00 | 3.74 | 3.85 |
| | **Talk2Me** | **4.54** | **4.50** | **4.49** | 3.81 | **4.47** |

Table 3: **Ablation study of key components.** We report reconstruction and synchronization metrics. "–" denotes reconstruction metrics not applicable when RBPG generates poses autonomously.

| Methods | LSE-D↓ | LSE-C↑ | PSNR↑ | LPIPS↓ | MS-SSIM↑ | FID↓ | LMD↓ |
|---|---|---|---|---|---|---|---|
| w/o LPE, EG, RBPG | 9.853 | 4.948 | 25.912 | 0.036 | 0.946 | 16.018 | 5.280 |
| w/o LPE | 8.067 | 7.011 | - | - | - | - | - |
| w/o EG | 9.897 | 5.009 | - | - | - | - | - |
| w/o RBPG | 7.994 | 7.102 | **26.645** | **0.027** | **0.959** | **13.712** | **3.716** |
| **Talk2Me** | **7.949** | **7.162** | - | - | - | - | - |

### 5.3.3 USER STUDY.

To assess perceptual quality, we conduct a user study following the Mean Opinion Score (MOS) protocol across five criteria. As shown in Table 2, **Talk2Me** tops four of five aspects, especially audiovisual coherence and expressiveness, and delivers overall balanced, high-fidelity, audio-synchronized facial animation despite slightly lower raw image quality. For more details, please refer to Appendix A.4.

### 5.4 ABLATION STUDY.

To assess the contribution of each component, we conduct an ablation study (Table 3). Removing any module degrades performance, confirming its necessity. Removing LPE affects geometric deformation, which in turn reduces lip-sync accuracy, highlighting its role in audio–visual alignment. Excluding EG causes larger drops, underscoring the importance of the fusion design for temporal consistency. The comparison between rows 1 and 4 further indicates that LPE and EG enhance image quality, demonstrating the effectiveness of learnable spatial encoding and audio–expression alignment. We further compare LPE with the conventional sinusoidal encoding from multiple perspectives, please refer to Appendix A.2.2 for details. In addition, we further investigate the impact of each loss term on model performance, with more details provided in Appendix A.2.3.

## 6 CONCLUSION

In this paper, we present Talk2Me, a high-fidelity and controllable audio-driven avatar framework that advances realistic talking head synthesis. Built upon 3D Gaussian Splatting, Talk2Me introduces a learnable spatial encoding and a modified region-weighted mechanism to preserve facial detail and structure. Our Expression Generator and Retrieval-Based Pose Generator jointly model audio–expression–pose correlations, enabling synchronized lip motion, expressive facial dynamics, and natural head movement. Extensive evaluations demonstrate that Talk2Me achieves superior performance across photorealism, synchronization accuracy, and motion expressiveness, offering a robust solution for audio-driven avatar generation.

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

# A APPENDIX

## A.1 DATASET AND BASELINE DESCRIPTIONS

### A.1.1 DATASET DETAILS

Our main experiments are conducted on a selected subset of the HDTF dataset (Zhang et al., 2021b), a high-resolution talking-face corpus commonly used in prior work. The full dataset consists of 362 video clips (about 15.8 hours in total), from which we choose representative samples based on speaking clarity, pose stability, and identity diversity. All selected videos are center-cropped and resized to $512 \times 512$ resolution, with a fixed frame rate of 25 FPS. Background regions are extracted using the matting technique proposed in (Lin et al., 2022). To evaluate cross-lingual generalization, we further collect 25 Mandarin-speaking video clips featuring a wide range of speakers. These videos undergo the same preprocessing steps as the HDTF subset.

To supplement our visual comparisons, we also include a few publicly available video samples from prior works such as GeneFace (Ye et al., 2023), AD-NeRF (Guo et al., 2021), and ER-NeRF (Li et al., 2023). These samples are used for qualitative demonstration only and are not involved in model any quantitative evaluation.

### A.1.2 BASELINE METHODS

To ensure a comprehensive and fair evaluation, we compare our method against a wide range of state-of-the-art talking-face synthesis models spanning different representation paradigms. Specifically, our baselines include GAN-based methods such as Wav2Lip (Prajwal et al., 2020), VideoReTalking (Cheng et al., 2022), and IP-LAP (Zhong et al., 2023), which focus on generating realistic lip-synced facial motions. We also include NeRF-based approaches like ER-NeRF (Li et al., 2023), SyncTalk (Peng et al., 2024), Real3dPortrait (Ye et al., 2024b), and MimicTalk (Ye et al., 2024a), which leverage volumetric scene representations to improve 3D consistency and novel view rendering. Finally, we consider recent 3D Gaussian-based methods, including GaussianTalker (Cho et al., 2024) and TalkingGaussian (Li et al., 2024), which explicitly model facial geometry with point-based representations for efficient and photorealistic synthesis. These baselines cover both traditional and emerging paradigms, enabling a thorough comparison across fidelity and controllability.

**Wav2Lip.**

This method focuses on generating talking-face videos with accurate lip synchronization for arbitrary identities under unconstrained conditions. It employs a fixed, pre-trained lip-sync discriminator (based on SyncNet (Chung & Zisserman, 2016)) as a strong supervisory signal to guide the generator, avoiding adversarial training. This work also introduces new evaluation benchmarks and two metrics—Lip Sync Error-Distance (LSE-D) and Lip Sync Error-Confidence (LSE-C)—to quantitatively measure performance. Human evaluation results show that the generated videos achieve synchronization quality comparable to real footage and are consistently preferred over those produced by previous methods.

**VideoReTalking.**

This method proposes a three-stage framework for audio-driven talking-head video editing in natural scenes: (1) generating a face video with a canonical expression; (2) producing temporally aligned lip movements from audio; and (3) enhancing realism via face restoration. Expression normalization uses a fixed template as pose reference to support accurate lip motion. Final outputs are refined by an identity-aware restoration network guided by a StyleGAN (Karras et al., 2019) prior. Without requiring identity-specific training, the system generalizes well to unseen speakers and performs robustly in the wild.

**IP-LAP.**

IP-LAP is a two-stage framework for audio-driven talking-face synthesis. It first employs a Transformer-based module to predict lip and jaw landmarks from speech, incorporating reference landmarks and prior poses for identity consistency. In the second stage, a rendering network generates face frames by warping multiple static reference images based on predicted motion, then fusing them with a masked frame and sketch.

**ER-NeRF.**

To enable high-fidelity audio-driven portrait synthesis, this method proposes a region-aware conditional neural radiance field that explicitly models the spatial contributions of different facial regions. A tri-plane hash representation decomposes the 3D space into three orthogonal 2D planes, improving rendering efficiency and reducing hash collisions by pruning empty regions. To capture fine-grained correlations between facial areas and speech signals, a region attention module applies cross-modal attention to produce region-aware conditioning features. In addition, a lightweight and efficient adaptive pose encoding maps complex head movements into spatial coordinates.

**SyncTalk.**

This NeRF-based framework is designed for audio-driven talking-head synthesis with an emphasis on spatiotemporal synchronization. It jointly addresses identity preservation, lip synchronization, facial expression control, and head pose alignment through a modular architecture comprising a facial synchronization controller, a head stabilization unit, and a portrait rendering module. The controller maps audio signals to dynamic facial features via an audio-visual encoder and animation mapper, enabling accurate lip-sync and expressive control. The stabilization unit smooths and aligns

head motion using tracked facial keypoints, while the rendering module corrects common NeRF artifacts to improve visual fidelity.

**Real3dPortrait.**

This one-shot 3D talking-head synthesis framework generates photorealistic and animatable portraits from a single input image. It addresses challenges in identity preservation, expression fidelity, and head–torso coordination by integrating several key components. A pre-trained image-to-plane (I2P) model encodes strong 3D priors to reconstruct facial geometry, while a motion adapter modulates animation behavior based on input conditions. To enable natural torso motion and background flexibility, the system incorporates a head–torso–background super-resolution (HTB-SR) module. In audio-driven scenarios, it uses a general audio-to-motion (A2M) model to generate synchronized facial animation for previously unseen identities. Together, these components enable high-quality, identity-consistent 3D portrait synthesis in a one-shot setting.

**MimicTalk.**

MimicTalk enables personalized and expressive 3D talking-face generation by rapidly adapting to target identities and producing individualized facial motion from audio. Instead of relying on identity-specific training from scratch, it builds upon a generalized NeRF-based model and introduces a hybrid adaptation pipeline that disentangles and learns both static appearance and dynamic motion characteristics from a few input samples. This approach significantly reduces training time—reportedly achieving adaptation in just minutes, over 47× faster than conventional methods. Additionally, the framework integrates an intrinsically stylized audio-to-motion module (ICS-A2M), which mimics the conversational style of a reference video while preserving the content integrity of the driving audio.

**GaussianTalker.**

Leveraging the high rendering efficiency of 3D Gaussian Splatting (3DGS), GaussianTalker enables real-time, audio-driven talking-head synthesis with controllable head pose. It constructs a single 3DGS representation of the head and introduces a mechanism to deform the Gaussians synchronously with the input speech. To enable controllability, Gaussian attributes are embedded into a shared implicit feature space, which interacts with audio features to produce temporally aligned deformation signals. A spatial audio-attention module further refines these embeddings to predict per-Gaussian offset trajectories. By enforcing local coherence and leveraging spatial priors, the system achieves stable manipulation of large sets of Gaussians with complex attributes.

**TalkingGaussian.**

To address the blurriness often observed in dynamic facial regions of NeRF-based models, TalkingGaussian introduces a structure-persistent deformation field grounded in 3D Gaussian Splatting. Smooth and continuous deformations are applied to stable Gaussian primitives, avoiding abrupt appearance transitions and resulting in clearer, more accurate head synthesis. To further enhance realism and synchronization, a motion decoupling module is incorporated to disentangle facial and oral dynamics, which simplifies training and improves lip-sync fidelity.

## A.2    ADDITIONAL EXPERIMENT

### A.2.1    GENERALIZATION ABILITY

To further evaluate the generalization and robustness of our method, we conduct additional experiments on a set of supplementary data samples. We compare the proposed method against all baseline methods, and report the qualitative visual performance of each method.

In Figures 4 and 5, we present the lip-sync performance across consecutive frames on the supplementary dataset. In terms of *lip-sync accuracy*, our method exhibits almost identical results to the ground truth, demonstrating the strong capability of **Talk2Me** in audio-driven talking head synthesis. Compared to the traditional GAN-based method **Wav2Lip**, our approach delivers superior image quality. As shown in Figures 4 and 5, Wav2Lip often produces noticeable bounding box artifacts around the mouth region. Moreover, it is visually evident that **VideoReTalking** performs

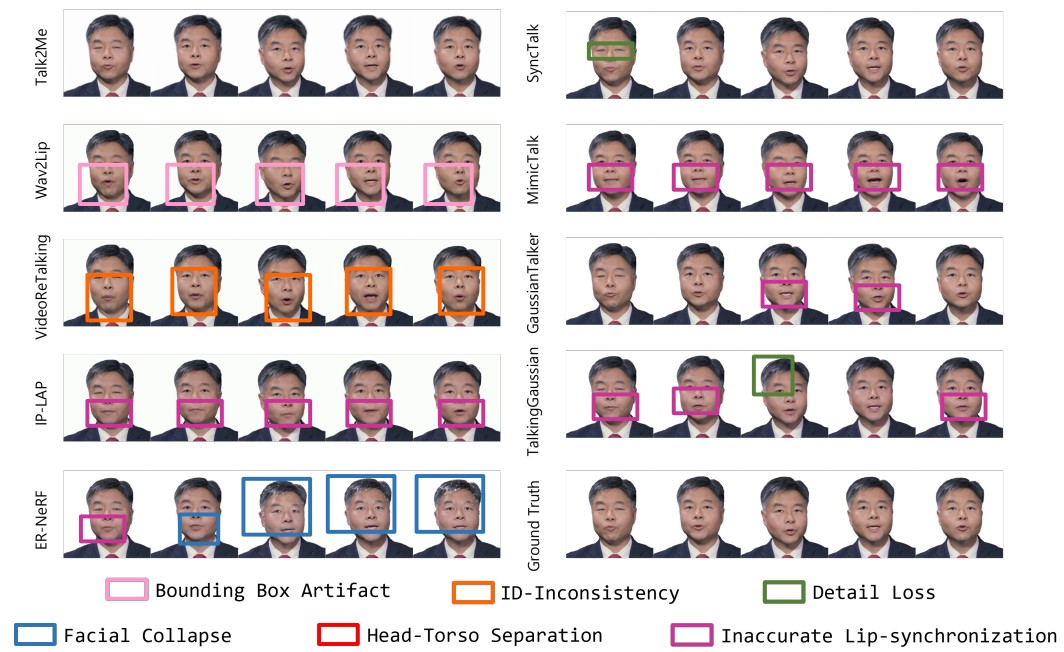

Figure 4: **Qualitative comparison of lip-sync performance on the supplementary dataset (Speaker A).** *Talk2Me* achieves superior lip-sync accuracy and identity preservation, with results close to the ground truth and fewer artifacts compared to prior GAN-, NeRF-, and 3DGS-based methods. Please zoom in for more details.

poorly in preserving speaker identity — the mouth appears to belong to a different person rather than the target identity. In contrast, **Talk2Me** maintains identity consistency remarkably well.

Compared to NeRF-based methods, our approach demonstrates significantly better preservation of fine-grained details and a more coherent connection between the head and torso. We attribute the head-torso discontinuity observed in NeRF-based methods to the *separate treatment* of these regions during the training phase.

Furthermore, in comparison to the other two 3DGS-based methods, the proposed method demonstrates more stable detail preservation and lip-sync performance across temporally continuous frames. For more intuitive and detailed comparisons, please refer to the supplementary videos provided, where we manually annotate the artifact-prone regions to better illustrate the differences.

### A.2.2 DETAILED VISUALIZATION RESULTS OF LPE ABLATION

In this section, to more intuitively and thoroughly validate the effectiveness of LPE, we provide additional visualization-based ablation results from three perspectives: the geometric quality of Mesh Head, the preservation of pupil illumination in Gaussian Head, and the preservation of eye details in Gaussian Head. These experiments highlight how LPE impacts generation quality from different aspects, further demonstrating its advantages in maintaining geometric consistency and detail fidelity.

**Geometric quality of Mesh Head**

Figure 6 illustrates the geometric quality of Mesh Head at different initialization steps. Without LPE, the Mesh Head exhibits noticeable unreasonable geometric structures, which persist throughout optimization. In contrast, with LPE, Talk2Me learns more stable and coherent geometric formations, and the rendered images at corresponding steps demonstrate significantly better quality. This indicates that LPE not only enhances positional encoding, but also provides stronger adaptability to finer

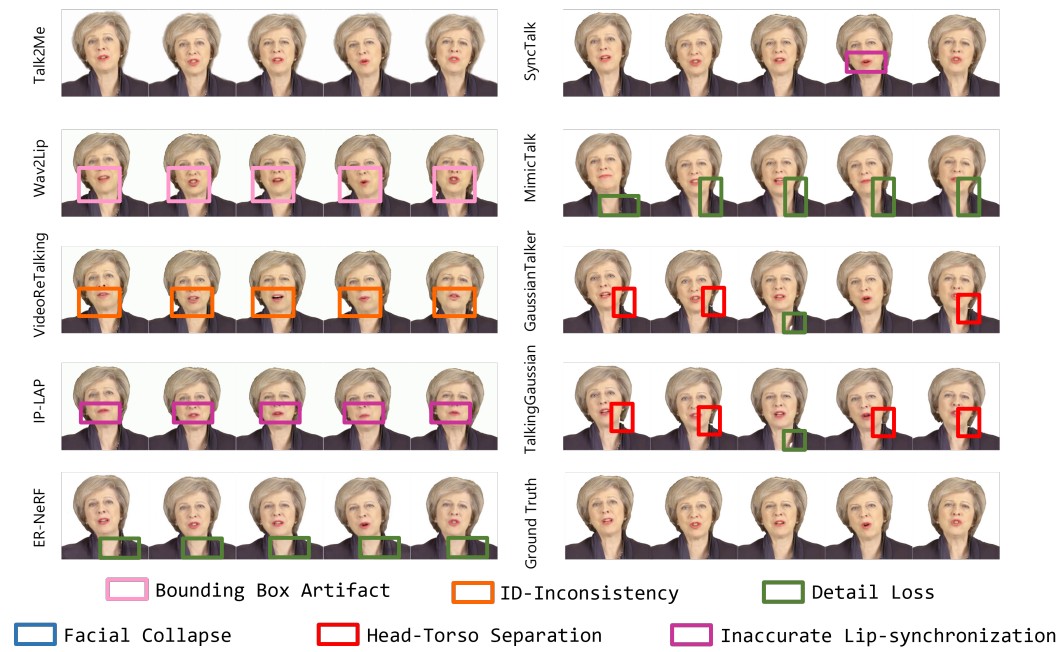

Figure 5: **Qualitative comparison of lip-sync performance on the supplementary dataset (Speaker B).** *Talk2Me* achieves superior lip-sync accuracy and identity preservation, with results close to the ground truth and fewer artifacts compared to prior GAN-, NeRF-, and 3DGS-based methods. Please zoom in for more details.

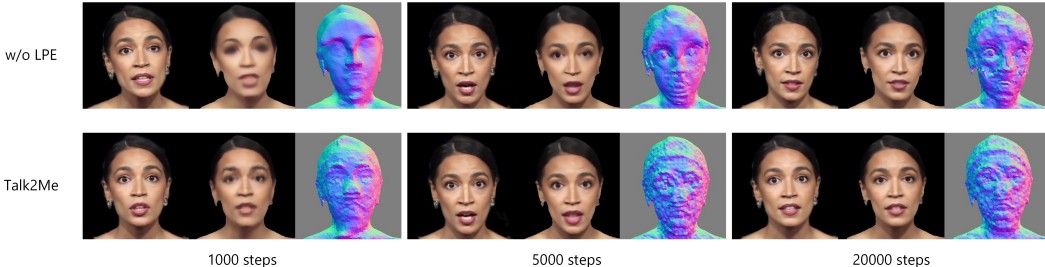

Figure 6: **Visualization of Mesh Head geometry at different training steps with and without LPE.** Without LPE, unreasonable geometric structures persist throughout optimization, while LPE leads to more stable geometry and higher-quality renderings.

local deformations. On top of the general geometric patterns captured by standard PE, LPE performs adaptive optimization, leading to superior structural fitting and more reliable reconstruction quality.

**Preservation of Pupil Illumination**

To examine the role of LPE in preserving eye illumination, we compare Gaussian Head renderings with and without LPE, as shown in Figure 7. With LPE, Talk2Me better preserves illumination in the eye region, resulting in higher brightness and greater visual fidelity. In contrast, without LPE, the ability to retain illumination details decreases, making the eyes appear darker and less bright, which leads to local detail loss and degraded visual fidelity. These results demonstrate that LPE

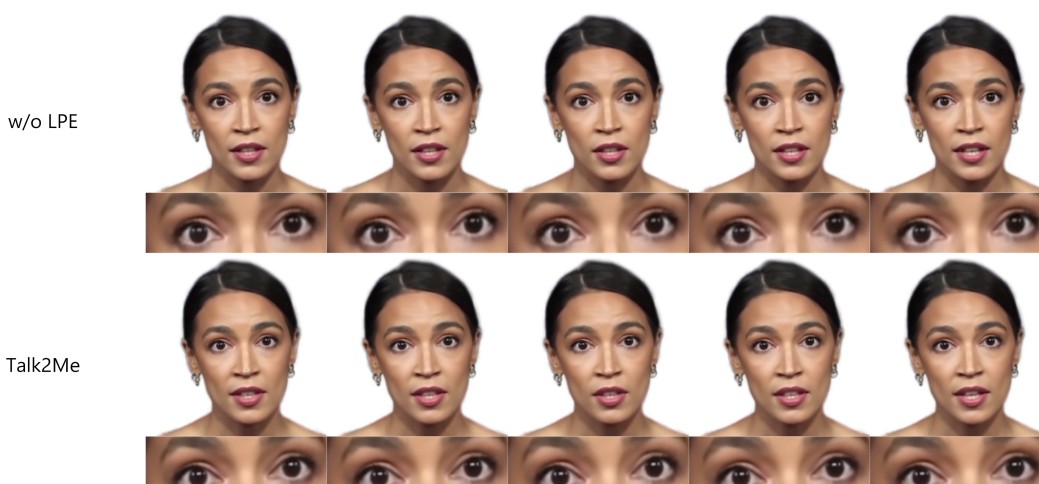

Figure 7: **Comparison of Gaussian Head renderings with and without LPE.** Without LPE, the ability to preserve illumination details in the eye region decreases, making the overall appearance darker and less bright. In contrast, LPE better maintains eye-region lighting, resulting in brighter and more faithful visual appearance. Please zoom in for detail.

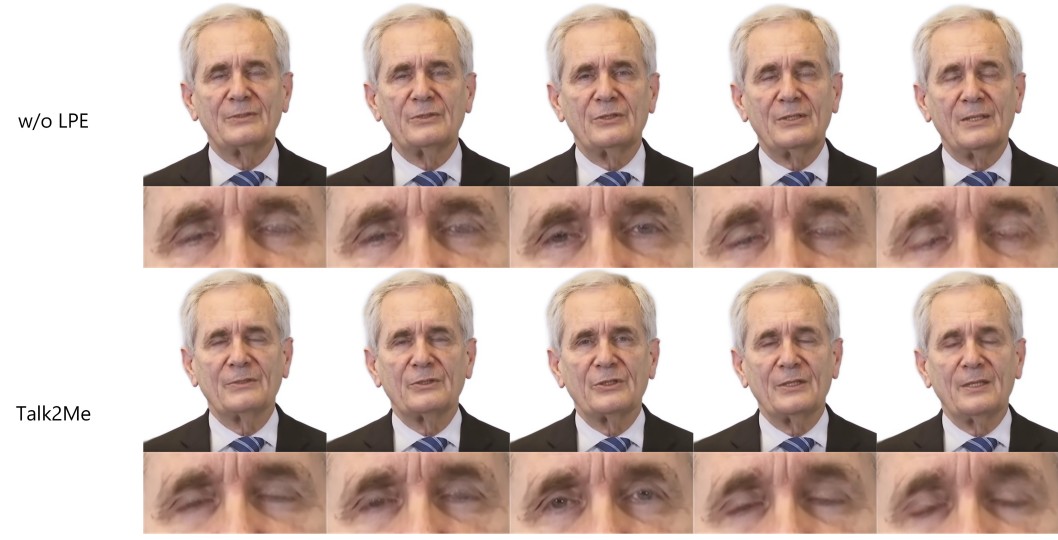

Figure 8: **Visualization of eye detail preservation in Gaussian Head renderings.** Without LPE, noticeable artifacts emerge in the eye region during blinking, whereas LPE enables Talk2Me to capture fine-grained dynamics with stable geometry and improved rendering quality.

significantly enhances the model's ability to capture subtle lighting variations, thereby improving the realism of the generated results.

**Preservation of Eye Details**

Furthermore, to evaluate the effectiveness of LPE in detail preservation, we compare Gaussian Head renderings with and without LPE, as shown in Figure 8. Without LPE, the eye region exhibits noticeable artifacts, leading to unstable geometry and loss of fine details. In contrast, with LPE, Talk2Me better captures subtle dynamics such as blinking, maintains stable geometric structures, and significantly improves overall rendering quality. These results further demonstrate the critical role of LPE in high-fidelity detail modeling.

### A.2.3 Ablation on Training Objectives

Table 4: **Ablation study on loss terms for Gaussian Avatar training. Bold** and underlined indicate the best and second-best results, respectively.

| Loss Term | PSNR↑ | LPIPS↓ | MS-SSIM↑ | FID↓ |
|---|---|---|---|---|
| $L_1$ | 28.867 | 0.054 | 0.975 | 0.218 |
| $L_1 + L_p$ | 28.658 | 0.045 | 0.973 | 0.113 |
| $L_1 + L_p + L_s$ | 29.724 | 0.042 | 0.977 | 0.112 |
| $L_1 + L_p + L_s + L_a$ | **29.966** | **0.038** | **0.978** | **0.014** |

Table 5: **Ablation study on loss terms for RBPG training. Bold** and underlined indicate the best and second-best results, respectively.

| Loss Term | LSE-D↓ | LSE-C↑ |
|---|---|---|
| $L_{gan} + L_{vel}$ | - | - |
| $L_{vel} + L_{rec}$ | 8.282 | 7.088 |
| $L_{gan} + L_{rec}$ | 8.254 | 6.844 |
| $L_{gan} + L_{rec} + L_{vel}$ | **7.949** | **7.162** |

To assess the contribution of each loss, we run ablations on **Gaussian Avatar** and **RBPG** (Eq. 10, Eq. 12); results are reported in Table 4 and Table 5. For Gaussian Avatar, adding the perceptual loss $L_p$ to $L_1$ improves perceptual quality: LPIPS changes from 0.054 to 0.045 and FID from 0.218 to 0.113, while PSNR and MS-SSIM decrease slightly. Introducing the SSIM loss $L_s$ then recovers fidelity, raising PSNR and MS-SSIM to 29.724 and 0.977, and further improving perceptual metrics (LPIPS = 0.042, FID = 0.112). Finally, adding the adversarial term $L_a$ provides the largest perceptual gain and the best overall accuracy (PSNR = 29.966, MS-SSIM = 0.978; LPIPS = 0.038, FID = 0.014).

For RBPG, note that removing the reconstruction loss prevents the model from generating plausible poses, and thus no results are reported in this case. Adding the velocity smoothness loss substantially improves temporal coherence (reflected by a lower LSE-D), while the adversarial loss enhances realism and reduces motion artifacts. The full objective achieves the most favorable trade-off, reaching the lowest LSE-D and highest LSE-C, which demonstrates the necessity of jointly considering all components.

### A.3 Model Architecture and Implementation

In this section, we provide a more detailed description of the proposed **Expression Generator** and the **Pose Refiner**.

#### Expression Generator

In Figure 9, we present a more detailed processing pipeline inside the **Expression Generator (EG)**. First, the input audio is processed by *wav2vec* to extract initial audio features. Then, we leverage a pre-trained *Wav2Lip* model to obtain richer audio representations, which provide more lip-related information. These audio features are then fed into an *Audio Encoder*, which maps them into the latent space of expression features. Simultaneously, an *Expression Encoder* extracts facial style features from a reference frame.

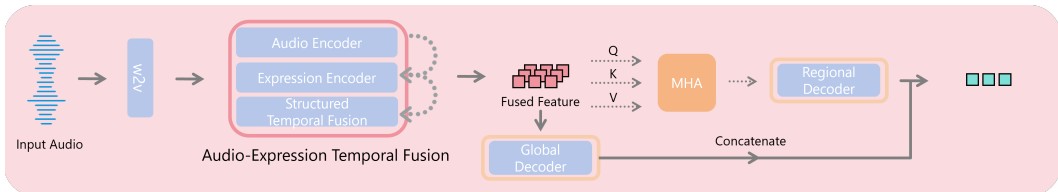

Figure 9: **Architecture of the Expression Generator (EG).** EG fuses audio and visual features via temporal modeling and outputs the predicted expression using both regional and global branches.

Then, a *Structured Temporal Fusion* module performs temporal modeling of the two modalities within the shared latent space, capturing cross-modal temporal correlations and outputting a fused feature representation.

After obtaining the fused feature, we design two branches to jointly predict the final expression. For the **AETF branch**, the fused feature is first processed by a *Multi-Head Attention* layer, followed by a *Regional Decoder* to generate a regional expression feature enriched with localized details. For the global branch, a more straightforward pathway is used to directly predict a *global expression feature* from the fused representation, which carries higher-level phonetic information. Finally, a lightweight, learnable weighting layer combines both branches as input and outputs the *Predicted Expression*.

ENCODER AND DECODER

All the encoders and decoders introduced in EG share the same lightweight architecture, as illustrated in Figure 10. This architecture consists of a few linear layers combined with *ReLU* activation function. Despite its simplicity, the design achieves superior performance while maintaining low computational overhead.

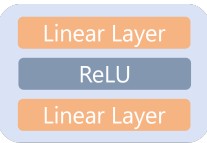

Figure 10: **Shared encoder-decoder architecture used in EG.** A lightweight design combining linear layers and ReLU, achieving efficiency without sacrificing performance.

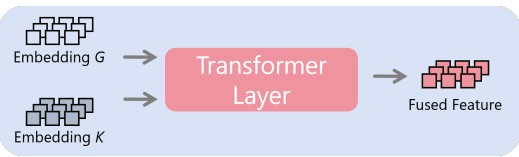

Figure 11: **STF module for multimodal fusion.** Audio and expression features are fused via gating, K-product interaction, and a Transformer layer to enhance temporal modeling.

STRUCTURED TEMPORAL FUSION.

The **STF** module within EG is illustrated in Figure 11. The features obtained from the *Audio Encoder* and *Expression Encoder* are first processed by a *feature gate* and a *K-product fusion* module, resulting in *Embedding G* and *Embedding K*, respectively. These embeddings are then fed into a *Transformer Layer*, which effectively aggregates expression and audio features, thereby enriching the temporal correlation information within the generated fused feature.

Figure 12: **Pose Refiner (PR).** Architecture of the PR module in RBPG. Pose, expression, and audio features are projected via linear layers. Pose and expression embeddings are fused using Kronecker Fusion, then combined with the audio embedding and processed by a Transformer layer. A final linear layer outputs the predicted pose.

POSE REFINER

The architecture of the **Pose Refiner** within the **Retrieval-Based Pose Generator (RBPG)** is shown in Figure 12. The retrieved pose, expression feature, and audio feature are each passed through individual linear layers to be projected into a shared latent space. Subsequently, a *Kronecker Fusion* layer takes the *pose embedding* and *expression embedding* as input to further model the relationship between facial expression and head pose. After this operation, the resulting *fused embedding*, along with the *audio embedding*, is fed into a *Transformer Layer*. The output is then passed through a final linear layer to produce smooth and natural head poses.

## A.4 USER STUDY DESIGN AND QUESTIONNAIRE

### A.4.1 OVERVIEW

To assess the perceptual quality of the generated talking head videos, we conduct a user study to evaluate the perceptual quality of the generated talking head videos following the standard Mean Opinion Score (MOS) protocol. The evaluation focuses on five key aspects:

- **Lip-sync accuracy (Lip)**: Alignment between lip movements and spoken audio.

- **Expression synchronization (Exp)**: Temporal and semantic consistency between facial expressions and the speech content.

- **Pose synchronization (Pose)**: Naturalness and coherence of head movement in response to speech.

- **Image quality (Img)**: Visual fidelity, clarity, and absence of artifacts.

- **Video realness (Vid)**: Overall realism of the video, including identity consistency and expressiveness.

Each criterion is rated on a 5-point Likert scale, where 1 indicates "poor" and 5 indicates "excellent." Participants watch anonymized video clips generated by various methods, including GAN-based, NeRF-based, 3DGS-based baselines, and our proposed **Talk2Me** model.

### A.4.2 STUDY SETUP

We conduct an anonymous online survey and obtain 120 valid responses. Each participant views a randomized set of anonymized video clips, each lasting 30–60 seconds, to reduce ordering effects and bias. Participants are asked to watch and rate the videos according to five specific criteria. They may replay each video as many times as needed before submitting their responses. No time limits are imposed, and participants complete the questionnaire at their own convenience.

The video dataset comprises a diverse set of speakers and utterances to mitigate potential bias related to speaker identity or language content. All participants are bilingual in English and Chinese.

### A.4.3 Ethics and Consent

All participants provide informed consent before participating in the study. They are informed about the purpose of the study, data privacy protection, and their right to withdraw at any time without consequence. No personal or identifiable data is collected.

The study involves non-sensitive, anonymous video content, while adhering to standard ethical guidelines for human-subject research.

### A.4.4 Questionnaire

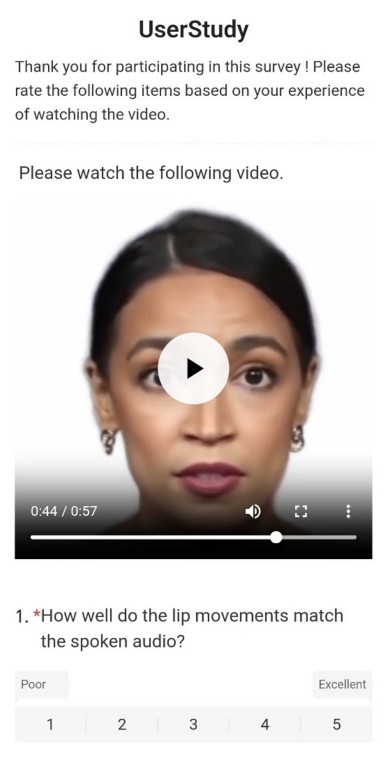
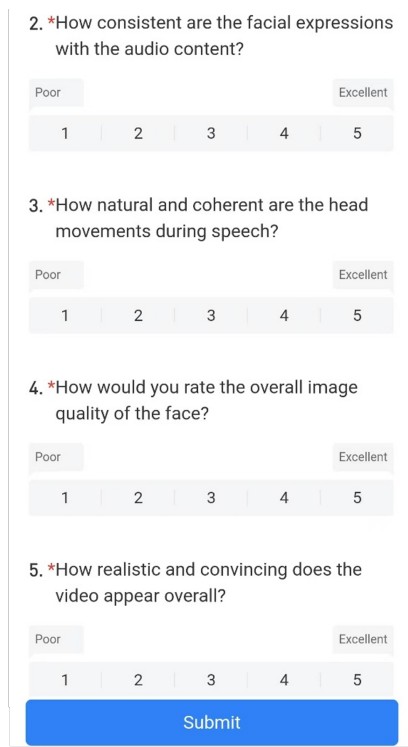

Figure 13: **User Study Questionnaire.** Participants rate generated videos on five aspects using a 5-point Likert scale: lip-sync accuracy (Lip), expression synchronization (Exp), pose synchronization (Pose), image quality (Img), and video realness (Vid).

As shown in Figure 13, each participant answers the following five questions after watching each video clip:

1. How well do the lip movements match the spoken audio? (**Lip-sync accuracy**)
2. How consistent are the facial expressions with the audio content? (**Expression synchronization**)
3. How natural and coherent are the head movements during speech? (**Pose synchronization**)
4. How would you rate the overall image quality of the face? (**Image quality**)
5. How realistic and convincing does the video appear overall? (**Video realness**)

All responses are collected digitally, and we compute the average MOS scores for each method and evaluation criterion. The final results are summarized in Table 2, where the best and second-best scores are shown in bold and underlined, respectively.

### A.4.5 Analysis and Results

As shown in Table 2, **Talk2Me** achieves the highest scores in four out of five evaluation aspects, and outperforms recent GAN-, NeRF-, and 3DGS-based methods. The method shows clear advantages

in audiovisual coherence and expressiveness. Although it ranks slightly lower in raw image quality, it demonstrates balanced and robust performance overall, generating high-fidelity and controllable facial animation that remains well synchronized with the input audio.

## A.5 Large Language Model Usage

In the process of writing this paper, we use a large language model (LLM) solely to refine the phrasing and improve the clarity of the text. The LLM is not involved in the design of the method, the experiments, or the analysis of results.

## A.6 Reproducibility

We ensure reproducibility by providing clear distinctions between results and interpretations, and by reporting implementation details such as hyperparameters, evaluation metrics, and computing infrastructure. All external datasets used in this work are publicly available and properly cited. In addition, we construct a small Chinese dataset, which will also be released upon publication. While certain aspects such as random seed settings and statistical significance tests are not reported, the information and resources provided are sufficient to enable independent reproduction of our results.

## A.7 Ethical Considerations in Avatar Generation

Our proposed method can be applied to avatar generation, enabling more realistic and stable 3D reconstructions. While such technology has promising applications in areas like virtual communication and entertainment, it may also raise concerns if misused. We therefore call for the responsible and ethical use of this technology to ensure it benefits creative and human-centered applications without infringing on privacy or identity rights.

