# OpenReview forum: "Talk2Me: High-Fidelity and Controllable Audio-Driven Avatars with Gaussian Splatting"
_ICLR.cc/2026/Conference — ICLR 2026 Conference Withdrawn Submission_

### Official Review · Reviewer_FGpT · 2025-10-24

**Soundness:** 3
**Presentation:** 2
**Contribution:** 2
**Rating:** 4
**Confidence:** 5

**Summary:**

The paper introduces Talk2Me, a framework combining 3D Gaussian Splatting, Learnable Positional Encoding, and a retrieval-based pose generator to create high-fidelity, controllable audio-driven avatars with improved facial animation, head movement, and lip sync, outperforming existing methods in quality and naturalness.

**Strengths:**

- The authors present a comprehensive set of experiments evaluating both reconstruction quality and audio-driven synchronization, with comparison to multiple baselines (GAN, NeRF, 3DGS). The results indicate clear advantages over existing methods.
- The user study demonstrates a significant improvement in perceptual quality, especially in lip-sync accuracy and expression synchronization.
- The paper makes a notable contribution by including a Mandarin dataset, which addresses cross-lingual generalization and enhances the robustness of the model.

**Weaknesses:**

- While the framework performs well in synchronizing speech and motion, it ranks slightly lower in terms of raw image quality compared to certain other methods (e.g., PSNR and MS-SSIM). The paper should better justify the trade-off between fidelity and expressiveness in facial motion synthesis.
- While the framework is promising in terms of fidelity and controllability, no details are provided on the computational efficiency of the system. Specifically, how feasible is the framework for real-time applications or large-scale deployment? A more in-depth discussion on the computational cost and real-time performance would be beneficial.
- The paper suggests that the 3DGS method helps improve the consistency of facial geometry, but some visual artifacts (e.g., noticeable blinking issues, light preservation in the eye region) still remain, especially without LPE. Further improvement or clarification in these areas would improve the robustness of the method.
- Some figures, such as Figure 3, are too blurry to understand.

**Questions:**

- Could you provide more detailed comparisons with state-of-the-art real-time audio-driven avatar generation systems regarding computational efficiency and latency? How scalable is Talk2Me for large-scale, real-world applications or live environments?
- The paper relies heavily on LPE to enhance geometric accuracy and texture fidelity. How does this approach compare to traditional methods for improving high-frequency texture details, such as NeRF or GAN-based models, especially in highly dynamic settings?
- While Talk2Me demonstrates promising results with Mandarin speech, can it handle significant variations in accent, emotion, or speech clarity? How well does it generalize to other languages or noisy real-world audio data?
- Could you elaborate on the temporal synchronization process within the AETF module? How does it manage edge cases where the audio is unclear or there are sudden pauses in speech?

---

### Official Review · Reviewer_dpjk · 2025-10-27

**Soundness:** 2
**Presentation:** 1
**Contribution:** 1
**Rating:** 2
**Confidence:** 3

**Summary:**

Talk2Me is a 3DGS-based, audio-driven 3D talking-head generation framework.  It addresses two major challenges:
1. Low fidelity, including facial collapse and loss of fine-grained details;
2. Controllable facial and head dynamics, and the correlation modeling between the two.

To this end, the paper proposes several key components:
- A Learnable Positional Encoding (LPE) and a modified Region-Weighted Mechanism to achieve high-fidelity rendering;
- An Expression Generator (EG) equipped with an Audio-Expression Temporal Fusion (AETF) module for controllable and expressive facial motion;
- A Retrieval-Based Pose Generator (RBPG) combined with a Pose Refiner (PR) to achieve controllable and realistic head pose generation.

**Strengths:**

- The authors incorporate a wide range of sophisticated modules to enhance the realism and controllability of 3D talking-head generation.
- The Retrieval-Based Pose Generator (RBPG) is an interesting contribution, leveraging an audio–pose database to retrieve and model semantically related head movements corresponding to audio cues.

**Weaknesses:**

1. (Major) The writing quality is poor. The paper reads like a technical report that lists many techniques and heuristics without clear logical flow. Existing methods and custom modifications are mixed together, leading to poor readability.

2. (Major) Although the method aims to improve fidelity and controllability, the demo videos still exhibit severe jitter, artifacts, and unsynchronized lip movements, suggesting that the proposed modules may not effectively address these issues.

3. (Minor) Many descriptions are unclear or missing crucial details (see Questions below). Likewise, Fig. 1 (overview) is very confusing and fail to illustrate how the components interact or operate.

**Questions:**

- In Equation (2), what do $d_{\text{far}}$ and $d_{\text{near}}$ represent?

- In the original 3DGS formulation, there is no explicit variable $z$. What does $z$ denote in this paper? In Equation (7), does $z$ correspond to some attribute of Gaussian?

- In Section 4.3.1, what is the architecture of the Expression Encoder? What is its role in the overall pipeline? Additionally, in line 257, should the audio encoder be Wav2Vec instead of Wav2Lip (possible typo)?

- In Section 4.3.1, what exactly is the global decoder in the Region-Aware Attention Mechanism? What are its inputs and outputs, and how are global and local features fused? Could the authors clarify this with a more detailed figure or explanation?

- What does the dual-pathway specifically refer to? Does it indicate the two decoders for global and local feature processing?

- Could the authors provide an ablation study for the Pose Refiner (PR), comparing the retrieved poses and the refined poses to demonstrate its effectiveness?

---

### Official Review · Reviewer_CYo4 · 2025-10-30

**Soundness:** 3
**Presentation:** 3
**Contribution:** 2
**Rating:** 2
**Confidence:** 1

**Summary:**

This paper proposes to address the challenges of low fidelity and limited controllability in audio-driven avatars by identifying key limitations in existing methods, such as reliance on traditional sinusoidal positional encoding leading to facial collapse and loss of fine details, as well as temporal misalignment and independent treatment of expression and head pose causing lip-sync errors and rigid motion. To overcome these issues, the authors present a 3DGS-based framework aimed at enhancing visual fidelity and enabling controllable facial and head dynamics. The proposed method incorporates Learnable Positional Encoding and a modified Region-Weighted Mechanism to address facial collapse and improve detail preservation. It also integrates the Eye Aspect Ratio feature for fine-grained blinking modulation and ensures identity consistency through 3DGS's inherent modeling capability. For expression controllability, an Expression Generator with an Audio-Expression Temporal Fusion module is proposed, enabling accurate lip synchronization and smooth expression transitions. For pose controllability, a Retrieval-Based Pose Generator and a Pose Refiner are introduced to generate natural and expressive head movements. The authors curate a Mandarin video dataset to assess cross-lingual generalization. Extensive evaluations on both English and Mandarin datasets demonstrate that Talk2Me achieves superior generation quality, synchronization accuracy, motion coherence, and expression controllability compared to existing methods. The main contributions include the presentation of Talk2Me, enhancements to 3D Gaussian Splatting, and the proposal of EG and RBPG for improved facial expression and head motion control.

**Strengths:**

The method is presented in a relatively clear manner, and the proposed approach is technically sound. The experimental validation is comprehensive, encompassing qualitative and quantitative comparative experiments, user studies, and ablation studies. Based on the presented results, the proposed method brings about certain performance improvements.

**Weaknesses:**

The novelty of the work is somewhat limited. Although the authors introduce multiple modules, they all lack innovation. Specifically, the region-weighted mechanism and learnable positional encoding are rather trivial methods, and the retrieval-based pose generator is more of an engineering-oriented post-processing step. I do not perceive much algorithmic innovation in these components.
In terms of experiments, the authors' comparative experiments are all conducted on a white background. Moreover, when comparing with other algorithms, it seems that the white backgrounds of other algorithms have been removed, which leads to a jagged edge issue around the human figures. In fact, many previous methods generate the human figure along with the background. The authors' approach of removing the background for comparison might create an unfavorable condition for other methods. I hope the authors can explain the motivation behind removing the background.
For person-specific 2D talking faces, there is actually limited room for improvement. From both quantitative and qualitative comparisons, the enhancements brought by the proposed method are quite marginal.

**Questions:**

I am particularly concerned about the issue of background removal mentioned in the weaknesses section, as I believe it may lead to unfair comparisons. Additionally, I would like to know whether the proposed method struggles to handle complex backgrounds, and I hope the authors can reply to this issue.

---

### Official Review · Reviewer_nyaT · 2025-10-31

**Soundness:** 3
**Presentation:** 3
**Contribution:** 2
**Rating:** 4
**Confidence:** 4

**Summary:**

This paper presents Talk2Me to generate controllable audio-driven 3D avatars built on 3D Gaussian Splatting. The authors propose to address limitations in previous talking-head methods including the lack of fine-grained facial details and limited controllability of facial expression and head pose. Specifically, they enhance 3DGS with a Learnable Positional Encoding (LPE) and a modified region-weighted deformation mechanism for accurate deformation and fine facial structures. Moreover, they introduce an Expression Generator with an Audio-Expression Temporal Fusion module (AETF) to model temporal correlations between speech and expression and a Retrieval-Based Pose Generator (RBPG) with a pose refinement strategy to improve pose controllability. The method is trained on HDTF dataset and further evaluated on a Mandarin video dataset to prove the cross-lingual generalization.

**Strengths:**

The paper aims to address challenges in existing audio-driven 3D facial animation methods, including facial-torso artifacts, unnatural motion, and audio–expression inconsistency. It combines learnable spatial encoding, region-aware deformation, and audio-conditioned expression and pose modeling into a unified 3DGS framework and shows comparisons with prior GAN-based, Nerf-based, and 3DGS-based baselines. The evaluations together with perceptual user study show certain improvement in the animation quality and lip-synchronization. In terms of clarity, the paper is overall easy to follow, with detailed architectural descriptions and ablation study that validates the design decisions. The implementation details and code also enhances reproducibility.

**Weaknesses:**

1. The improvements shown in the paper are incremental, as comparisons are demonstrated under reconstruction setting, which means they all have same poses, eyeblinks etc. And the expression variation is small, making it hard to evaluate the claimed gains in motion controllability and naturalness.
2. Some details are also missing: the paper introduces a Mandarin monocular dataset, but provides only brief information and it would also be helpful to compare performance differences between the HDTF test set and the Mandarin set to support the cross-lingual generalization claim. Moreover, the training data processing including expression and pose coefficients definitions, and extraction are missing.
3. Lack of results: there is no video ablation visualization and comparisons of head-pose generation against SOTA methods. The current experiments focus on reconstruction quality rather than demonstrating more motion-control advantages.

**Questions:**

1. Is the model person-specific or generalizable to unseen identities? If it is person-specific, how much data per subject is required, and is the performance sensitive to data length and quality?
2. The video results show relatively small head movements and limited expression variation (except lip motion). Is this due to limited motion in the training data or does the model implicitly bias toward average motion patterns? It would be helpful to show more examples with expressive motion.

---

### Note · Authors · 2025-11-21

I have read and agree with the venue's withdrawal policy on behalf of myself and my co-authors.